# Abnormal Cellular Phenotypes Induced by Three *TMPO*/LAP2 Variants Identified in Men with Cardiomyopathies

**DOI:** 10.3390/cells12020337

**Published:** 2023-01-16

**Authors:** Nathalie Vadrot, Flavie Ader, Maryline Moulin, Marie Merlant, Françoise Chapon, Estelle Gandjbakhch, Fabien Labombarda, Pascale Maragnes, Patricia Réant, Caroline Rooryck, Vincent Probst, Erwan Donal, Pascale Richard, Ana Ferreiro, Brigitte Buendia

**Affiliations:** 1Basic and Translational Myology Laboratory, Université Paris Cité, BFA, UMR 8251, CNRS, F-75013 Paris, France; 2APHP—Sorbonne Université, Unité Fonctionnelle de Cardiogénétique et Myogénétique Moléculaire, Service de Biochimie Métabolique, HU Pitié Salpêtrière—Charles Foix, F-75013 Paris, France; 3INSERM, UMR_S 1166, Sorbonne Université, F-75005 Paris, France; 4Faculté de Pharmacie Paris Descartes, Département 3, Université Paris Cité, F-75006 Paris, France; 5Service d’anatomopathologie, CHU de Caen, F-14000 Caen, France; 6Département de cardiologie, APHP—Sorbonne Université, HU Pitié Salpêtrière- Charles Foix, F-75610 Paris, France; 7Service de Cardiologie, CHU de Caen, Université de Caen Normandie, F-14000 Caen, France; 8Cardiologie pédiatrique, Service de pédiatrie, CHU de Caen, F-14000 Caen, France; 9Service de Cardiologie, Hôpital Haut Lévêque, CHU de Bordeaux, INSERM 1045, Université de Bordeaux, F-33000 Bordeaux, France; 10Service de Génétique Médicale, CHU Bordeaux, F-33000 Bordeaux, France; 11Centre de référence des maladies rythmiques cardiaques, CHU de Nantes, F-44000 Nantes, France; 12Centre Cardio-Pneumologique, CHU de Rennes Hôpital de Pontchaillou, F-35000 Rennes, France; 13APHP, Centre de référence des Maladies Neuromusculaires, Institut de Myologie, Neuromyology Department, CHU Pitié Salpêtrière—Charles Foix, F-75013 Paris, France

**Keywords:** *TMPO* gene, LAP2 proteins, nuclear alterations, chromatin-associated proteins, E2F1 regulation, gender, cardiomyopathy

## Abstract

A single missense variant of the *TMPO*/LAP2α gene, encoding LAP2 proteins, has been associated with cardiomyopathy in two brothers. To further evaluate its role in cardiac muscle, we included *TMPO* in our cardiomyopathy diagnostic gene panel. A screening of ~5000 patients revealed three novel rare *TMPO* heterozygous variants in six males diagnosed with hypertrophic or dilated cardiomypathy. We identified in different cellular models that (1) the frameshift variant LAP2α p.(Gly395Glufs*11) induced haploinsufficiency, impeding cell proliferation and/or producing a truncated protein mislocalized in the cytoplasm; (2) the C-ter missense variant LAP2α p.(Ala240Thr) led to a reduced proximity events between LAP2α and the nucleosome binding protein HMGN5; and (3) the LEM-domain missense variant p.(Leu124Phe) decreased both associations of LAP2α/β with the chromatin-associated protein BAF and inhibition of the E2F1 transcription factor activity which is known to be dependent on Rb, partner of LAP2α. Additionally, the LAP2α expression was lower in the left ventricles of male mice compared to females. In conclusion, our study reveals distinct altered properties of LAP2 induced by these *TMPO*/LAP2 variants, leading to altered cell proliferation, chromatin structure or gene expression-regulation pathways, and suggests a potential sex-dependent role of LAP2 in myocardial function and disease.

## 1. Introduction

Mutations in genes encoding nuclear envelope proteins (lamins A/C and B, lamin B receptor, emerin, MAN1) and their partners (lamin-associated proteins (LAPs), barrier autointegration factor (BAF)) have been identified as a cause of several human diseases [1,2,3]. Strikingly, defects in the *LMNA* gene, that encodes lamins A/C, cause about a dozen of clinical disorders, including skeletal muscle pathologies, dilated cardiomyopathies, lipodystrophies, peripheral neuropathies and the rare Hutchinson–Gilford progeria syndrome [1]. Isolated cardiomyopathies, namely hypertrophic cardiomyopathy (HCM) and dilated cardiomyopathy (DCM) [4] are diseases of the cardiac muscle with a relatively high prevalence (1:500 for HCM and 1:3000 for DCM). DCM is characterized by ventricular dilatation and reduced myocardial systolic contractility with left ventricular ejection fraction (LVEF) below 40%, while HCM is characterized by left ventricular hypertrophy affecting preferentially the interventricular septum (>15 mm). HCM and DCM are often familial (20 to 40% cases) with a large genetic and allelic heterogeneity [5,6,7]. *LMNA* is the second most commonly mutated gene in DCM [8] but despite extensive gene analyses, the genetic defect for nearly half of the patients with familial cardiomyopathies is still unknown.

The hypothesis that defects of *TMPO,* encoding LAP2α,β proteins [9,10], could be associated with cardiomyopathy is attractive since LAP2α is a direct partner of lamins A/C [11]. LAP2 and lamins contribute to cell cycle regulation and genome organization/expression [12,13,14]. There are two major LAP2 isoforms, α and β, generated by alternative splicing. Both contain in their N-ter region, a LEM domain, shared with LEMD2, Emerin and MAN1 [15], which mediates interaction with the chromatin-associated protein BAF [16,17]. The two LAP2 isoforms differ by their C-ter region, which contains either pRb-, lamin A/C- and HMGN5-binding domains (LAP2α) or lamin B1- and HDAC3- binding domains (LAP2β) [18,19]. In mouse models, complete *TMPO* knockout leads to ventricular systolic dysfunction and late myocardial fibrosis exclusively in males, consequently to deregulated expressions of genes involved in cardiac remodeling and fibrosis [20]. In humans, only one *TMPO* variant (c.2068C>T, p.(Arg690Cys)) was described in two brothers affected by pure DCM without significant cardiac conduction problems [21]. However, this variant is now identified in the gnomAD population database with a Minor Allele Frequency (MAF) = 1.7% well above the pathogenic frequency limit of MAF < 0.01% defined in HCM, the most prevalent cardiomyopathy [22]. Thus, it is no longer unclear whether the p.(Arg690Cys) *TMPO* variant can cause DCM [21]. It is clear that this variant is a functional polymorphism, highly frequent in Latin American populations.

Here, we report (i) the clinical phenotype of six male patients diagnosed of cardiomyopathy (HCM or DCM), harboring three new *TMPO* variants and (ii) the cellular properties of these new *TMPO* variants, in comparison with the previously reported p.(Arg690Cys) variant [21]. In cells, we tested the relative expression level of these variants and their in situ proximity with two regulators of chromatin compaction, BAF and HMGN5, which bind the LEM domain of LAP2α and LAP2β, and the C-ter region specific of LAP2α, respectively [17,23]. We also tested the impact of LAP2α variants on E2F1 because LAP2α was shown to contribute to inhibiting the activity of the E2F1 transcription factor [24]. Of interest, while E2F1 is mainly known to regulate cell cycle progression, it also plays a role in metabolism pathways, whose deregulation is associated with disease, including cardiomyopathies [25]. Intrigued by the exclusivity of the development of heart disease in men harboring *TMPO*/LAP2 mutations (while the mothers who transmitted the mutation were asymptomatic), we investigated LAP2 expression levels in left ventricles from the hearts of male and female mice. The results suggest a sex-based difference in LAP2α expression. Altogether, our observations suggest a role of the three new rare *TMPO* variants identified in cardiomyopathy patients in altering the properties of LAP2 proteins, and subsequent dysfunction of partners playing a role in chromatin compaction and/or gene expression.

## 2. Materials and Methods

### 2.1. Cardiomyopathy Gene Panel Sequencing

DNA from ~5000 cardiomyopathy index cases was extracted from peripheral blood (Quiasymphony^®^, Qiagen, Düsseldorf, Germany) and sequenced on a targeted custom panel of coding sequences of 46 known genes (see Appendix A) involved in the different cardiomyopathies and their phenocopies [26]. This list includes the 4 exons of *TMPO* encoding the LAP2α isoform (Appendix A). A variant interpretation was performed accordingly with the American College of Medical Genetics and Genomics (ACMG) guidelines adapted in the context of cardiomyopathies. We evaluated the pathogenicity of each variant considering several parameters: (a) a frequency threshold <0.01% in gnomAD v2.1.1 database (URL: http://gnomad.broadinstitute.org/, accessed on 6 January 2023) [7]; (b) in silico predictions from multiple algorithms (for missense and splicing variants); (c) location of the variant in the gene and protein; (d) a careful literature review (HGMD Pro and PubMed); (e) functional studies and segregation analyses when available. The variants have been classified as certainly pathogenic (class 5), likely pathogenic (class 4), variant of unknown significance (class 3), likely benign (class 2) and benign (class 1). We have retained 3 *TMPO* variants classified as class 4 or 5. In addition, in order to exclude any other putative cause of genetic cardiomyopathy due to a variant in a gene not included in the panel, exome sequencing were performed (Twist Biosciences technology and IntegraGen Genomics (Evry, France)).

### 2.2. Cell Culture and Transfection

Mouse C2C12 myoblasts, rat H9C2 cardiomyocytes and Human embryonic kidney HEK293 cell lines were obtained from American Type Culture Collection. HEK293 fibroblasts were grown in MEM Eagle alpha modification plus 1% glutamine and 10% fetal bovine serum (FBS). C2C12 and H9C2 cells were grown in the DMEM medium containing high glucose, GlutaMAX^TM^ (Gibco) and 10% FBS. Primary cultures of human skin fibroblasts (patient 1 aged 73 years; healthy control aged 59 years) were grown in DMEM medium containing low glucose, GlutaMAX^TM^ (Gibco), 1% non-essential amino acids and 10% FBS. All cells were cultured in the presence of antibiotics (penicillin, streptomycin). For cell transfection, Lipofectamine^TM^ 3000 (Life Technologies) or XtremeGene9 (Roche) were used according to the manufacturer’s instructions. Cells were analyzed 24 h to 40 h after transfection.

### 2.3. Plasmids

pCMV6 vectors encoding human LAP2α WT (amino acids 1-694) and LAP2β WT (amino acids 1–454) tagged with myc-DDK in C-ter were obtained from OriGene (Cliniscience). The plasmids encoding the missense variants (p.(Leu124Phe), p.(Ala240Thr), p.(Arg690Cys) and the single nucleotide deletion variant p.(Gly395Glufs*11) were generated using the QuickChange Lightning site-directed mutagenesis kit (Agilent Technologies). An additional plasmid was generated for the deletion variant in order to escape nonsense-mediated mRNA decay. This plasmid encoding LAP2α p.(Gly395Glufs*11)-DDK was constructed by amplifying (PCR) the cDNA region of interest (bp 1 to 1213) using as a template the plasmid pCMV6 encoding p.(Gly395Glufs*11), and ligating into the Sgf1 and Not1 sites of a pCMV6 plasmid. pcDNA-3HA-BAF vectors encoding HA-BAF were generated by PCR amplification of the full-length cDNAs using as template a pEGFP vector encoding-WT BAF [27] followed by ligation of the amplified DNA into the EcoR1 and XBaI sites of pcDNA-3HA (Addgene). Primers are listed in the Online Appendix A. All sequences were verified by Sanger sequencing.

### 2.4. qRT-PCR

RNAs were purified from H9C2 cells transfected with the plasmids encoding LAP2α (3 independent experiments) and from human dermal fibroblasts at early passage (p3) using the Nucleospin RNA kit (Macherey Nagel, Dueren, Germany). Then, RNAs were transcribed into cDNAs using the First-strand cDNA synthesis kit for RT-PCR (Roche, Switzerland). LAP2α mRNA quantification was done by qRT-PCR using reference genes validated with the NormFinder algorithm (GAPDH or YWHAZ for rat H9C2 cells, and HPRT for human dermal fibroblasts). All primers are listed in Appendix A.

### 2.5. Immunofluorescence Methods

For immunofluorescence (IF), cells were fixed with paraformaldehyde 3% for 12 min at R.T, then processed as described [28]. Immunofluorescence was observed using the confocal microscope Zeiss LSM 700 at the Imaging Facility of BFA institute.

### 2.6. Cell Protein Extract Preparation and Protein Analysis by Western Blot

Whole-cell extracts were prepared by suspending cells in the Laemmli sample buffer and analyzed by immunoblot as described [29]. ImageJ was used to quantify ECL signals.

Alternatively, cell fractionation was performed following a protocol adapted from Mittnacht and Weinberg [30]. Briefly, cells in Petri dishes were extracted in a buffer (10 mM Hepes pH 7.9, 10 mM KCl, 1.5 mM MgCl_2_, 0.5 mM DTT and 0.4% Triton) complemented with protease inhibitors (complete EDTA free, Roche, Germany) and phosphatase inhibitors (PhosSTOP, Roche, Mannheim, Germany), for 10 min at 4 °C. Solubilized proteins were precipitated with 10% trichloroacetic acid (TCA) before resuspension in Laemmli buffer. Insolubilized proteins were directly suspended in Laemmli buffer.

### 2.7. Co-Immunoprecipitations

Non-transfected and transfected HEK293 cells expressing LAP2-DDK WT or L124F together with HA-BAF were extracted in lysis buffer containing 25 mM Tris-HCl pH 7.4, 137 mM NaCl, 2.5 mM KCl, 1% Triton and supplemented with a cocktail of protease inhibitors and phosphatase inhibitors (see above) for 30 min at 4 °C with gentle agitation. Then, extracts were sonicated, cleared by centrifugation and the total protein concentration of lysates was determined by BCA assay (Pierce/Thermo Scientific). Equal amounts of proteins were incubated with a mouse anti-HA antibody or mouse anti-GFP antibody (used as a negative control) for 1 h at 4 °C with end-over-end rotation, followed by incubation with protein G plus agarose beads (SantaCruz Biotechnology, Dallas, TX, USA) for an additional hour at 4 °C. After centrifugation, both the supernatants and pellets of immunoprecipitation were kept. The protein G plus agarose bead pellets were washed with a lysis buffer containing 0.1% Triton, and finally, resuspended in a Laemmli sample buffer. In parallel, the supernatants were used for the next incubation with Mouse anti-FLAG-M2 affinity gel (Sigma-Aldrich, St. Louis, MI, USA) in order to recover LAP2 proteins tagged with DDK. After 1.5 h at 4 °C with end-over-end rotation, and four washes with lysis buffer containing 0.1% Triton, proteins immunoprecipitated by the FLAG-affinity gel were solubilized in Laemmli sample buffer.

### 2.8. Antibodies

Antibodies used for IF and Western blots are listed in the Appendix A.

### 2.9. Reporter Assay

HEK293 cells were co-transfected with the E2F-luciferase reporter vectors (Cignal E2F reporter assay kit, Qiagen) and, either an empty vector, or vectors encoding LAP2 proteins alone or together with a vector (pcmv3) encoding 3HA-E2F1. Luciferase expression was quantified 40 h post-transfection, using the Dual-Luciferase Reporter Assay System (Promega) and data were normalized as described [29].

### 2.10. Proximity Ligation Assays (PLA)

Quantification of proximity events occurring between LAP2α-DDK WT or mutant and either BAF, Lamin A/C or HMGN5 was performed using PLA, as previously described [31,32]. Data were expressed in mean intensity values for the nuclear PLA signals observed on confocal microscope images and quantified with ImageJ.

### 2.11. Mouse Heart Sample Extract Preparation and Protein Analysis

Nine-week-old male and female C57Bl6J mice (Envigo) were euthanized by cervical dislocation and hearts were rapidly excised and weighed (n = 10 for each sex). Part of the left ventricle was frozen in liquid nitrogen for further biochemical determinations. All animal experimental procedures complied with directive 2010/63/EU of the European Parliament on the protection of animals used for scientific purposes. Frozen tissue samples were homogenized at 4 °C using 1.4 mm ceramic beads combined with one 2.8 mm bead (Bertin precellys) in an ice-cold RIPA buffer containing an antiprotease cocktail set. Around 15 to 20 µg of protein extract was loaded per well in a gradient gel (4–20% acrylamide; Bio-Rad, France) and Western blots were performed (see above) in order to detect LAP2 proteins.

### 2.12. Statistical Analyses

Kruskall–Wallis, Mann–Whitney and One-way Anova tests were performed using GraphPad Prism7.

### 2.13. Collection of Biological Samples

An analysis of patient samples was performed in the context of the etiologic diagnosis of the disease in the standard care of patients. Molecular tests were done after the signature of an informed written consent approved by the local ethic committee of Pitié Salpêtrière Hospital according to the French Legislation (article R. 1131-4 du code de la santé publique). The patient’s DNAs were included in a collection of biological resources at APHP, Assistance Publique des hôpitaux, Paris, France. This collection was declared under the following number Biological resources n° DC 2009-957.

## 3. Results

### 3.1. Three Novel TMPO Variants Identified in Patients with Cardiomyopathies

An analysis of our cohort of ~5000 patients with cardiomyopathies with the targeted gene panel identified three heterozygous *TMPO*/LAP2α variants (NM_003276.2) with MAF ≤ 0.0027% and classified them according to ACMG guidelines (see Section 2) as pathogenic or likely pathogenic. For all patients, we confirmed the absence of additional mutations that could be classified as disease-causing in the major known cardiomyopathy genes (see Appendix A) as well as in exome sequencing.

A frameshift variant g.98927219del_c.1184del_p.(Gly395Glufs*11) was found in two unrelated patients diagnosed with HCM. This is a single nucleotide deletion in exon 4, the last LAP2α encoding exon and specific of this LAP2 isoform (Figure 1A,B). It is predicted to lead to a truncated protein devoid of the C-ter domains (which interact with HMGN5, pRb and lamin A/C [18]) and/or to haploinsufficiency.

Additionally, two missense variants were observed in DCM patients. One missense variant g.98926753G>A_c.718G>A_p.(Ala240Thr), identified in two brothers diagnosed with DCM, is located within a region specific to LAP2α, likely involved in HMGN5 binding [19] (Figure 1A,C). Another missense variant, g.98921754C>T_c.370C>T_p.(Leu124Phe) identified in two unrelated patients diagnosed with DCM, is located in exon 2, encoding a region which is common to LAP2α and β (Figure 1A,C). The mutated amino acid is in the LEM domain that binds BAF [16,17]. These two new missense variants were predicted as damaging (Provean), highly destabilizing and disease-associated (Strum method [33]).

### 3.2. Clinical Phenotype of Patients Harboring New TMPO Variants

#### 3.2.1. Patients Harboring the TMPO Frameshift Variant g.98927219del_c.1184del_p.(Gly395Glufs*11)

Patient 1 presented, at 58 years, with dyspnea leading to the diagnosis of asymmetric HCM with predominant septal hypertrophy and conserved ejection fraction (55%). Dyspnea progressed to class II-III severity as defined by the New York Heart Association (NYHA). Echocardiography and an MRI performed at 68 years revealed obstructive asymmetric hypertrophy localized within the basal antero-septal wall (Type I), 18 mm maximal wall thickness, outflow tract gradient = 200 mm Hg and systolic anterior movement. The left atrium was not enlarged. A 12-lead ECG (under diltiazem and atenolol) displayed sinus bradycardia (49 bpm), normal axis, normal PR interval, intraventricular conduction defect with atypical left bundle branch block pattern (QRS 125 ms) and normal QT duration. MRI showed no intra-myocardial fibrosis or late gadolinium enhancement. A syncope at 70 years with a high risk of sudden death led to defibrillator implantation. Drug treatment controlled dyspnea until further progression at 74 years, when a clear-cut atrio-ventricular block appeared. Patient 1 harbors the p.(Gly395Glufs*11) *TMPO* variant (Figure 1) and exome sequencing did not reveal any additional variants that could be involved in the development of cardiomyopathy. Other causes of HCM (hypertension, mitral abnormalities, intensive sport practice) were clinically excluded. DNA from relatives for family segregation studies was unavailable. His mother and father had no antecedent of cardiac defects and died at 87 and 92 years, respectively. His brother died at 40 years (sudden cardiac death), but no autopsy was performed.

Patient 2 presented with palpitations at 34 years, unveiling an asymmetric non-obstructive hypertrophic cardiomyopathy, with 16 mm maximum wall thickness at the basal antero-septal wall (Type I), intraseptal late gadolinium enhancement, and interventricular septum (IVS) fibrosis evidenced by echocardiography/MRI. Left atrium was not enlarged. The exercise test, 24 h Holter monitoring and ECG displayed no significant abnormality. Patient 2 harbors the p.(Gly395Glufs*11) *TMPO* variant (Figure 1), which was also carried by his mother (born in 1950) and a younger brother (born in 1977) whose cardiac evaluation has not revealed abnormalities so far. Exome sequencing performed on the index patient and his parents (family II) confirmed the maternal inheritance of the *TMPO* variant but did not reveal any additional pathogenic variant that could explain the cardiomyopathy.

#### 3.2.2. Clinical Phenotype of Patients Harboring the Missense Variant g.98926753G>A_c.718G>A_p.(Ala240Thr)

Patients 3 and 4 were two brothers diagnosed with DCM with severely reduced LVEF (15% and 25%) at ages 44 and 42 years, respectively. Both presented an incomplete left bundle branch block, and the older brother was recently implanted with a cardioverter defibrillator. Patients 3 and 4 harbor the p.(Ala240Thr) *TMPO* variant (Figure 1), also carried by their mother who was asymptomatic. In depth phenotypical studies are in progress.

#### 3.2.3. Clinical Phenotype of Patients Harboring the Missense Variant g.98921754C>T_c.370C>T_p.(Leu124Phe)

Patient 5, a baby with a birth weight of 3980 g, was diagnosed with dilated cardiomyopathy at age 3 weeks, after a syncopal episode. Metabolic, infectious and toxicological evaluations were negative. At 2.5 months, he developed acute pulmonary oedema following cardiac decompensation and required hospitalization and dobutamine treatment. Echocardiography showed ventricle dilatation with global hypokinesia (shortening fraction: 12%) and left atrial dilatation. The patient died in the course of this episode at 2.5 months. Macroscopic and histopathological analysis of postmortem ventricular myocardium showed myocardial disarray with hypertrophic cardiomyocytes and severe fibroelastosis. Both the mother and father had normal echocardiography and electrocardiogram at ages 35 and 41 years, respectively. This patient harbors the p.(Leu124Phe) *TMPO* variant (Figure 1), also carried by his mother who was asymptomatic. The exome sequencing in the baby’s DNA did not reveal any variants that could be considered as putatively pathogenic, inborn errors of metabolism or other known causes of neonatal cardiomyopathy.

Patient 6 presented with a complete atrioventricular block but normal echocardiography at age 54 years, requiring pace-maker implantation. Subsequently, he developed a dilated cardiomyopathy with LVEF = 40%. At 59 years, an asymptomatic episode of non-sustained ventricular tachycardia (190/min) was recorded. He harbors the p.(Leu124Phe) *TMPO* variant (Figure 1). No relatives were available for cardiac evaluation and genetic studies.

### 3.3. Sex and Cardiomyopathy Associated with TMPO Variants

In our study, we noticed that all patients with cardiomyopathy associated with *TMPO* variants were men. Furthermore, in families where sons with cardiomyopathy had inherited a *TMPO*/LAP2 variant from their mothers, the latter were asymptomatic. Interestingly, it was previously reported that mice depleted of LAP2α (LAP2α −/−) developed cardiac abnormalities (systolic dysfunction), but these were observed only in males (either young (10 weeks) or old (10 months)) and not in females [20]. Yet, the reason for such a difference was not known. We wondered whether the higher susceptibility of males to develop cardiac defects might be related to a distinct expression level of LAP2 proteins in the heart of males vs. females. Thus, we dissected the left ventricle (LV) from the hearts of control young mice (9 weeks) and extracted whole proteins. Quantitative analysis of Western blots incubated with anti-LAP2α antibodies revealed a reduced amount of LAP2α protein expressed in males vs. females ventricles, whatever the reference protein used (Red ponceau staining, GAPDH, tubulin or histone H4 (Figure 2A,B and Appendix A). Subsequent incubation of Western blots with *TMPO* antibodies that recognize all LAP2 isoforms (α, β, δ, γ), revealed that LAP2β-like isoforms are more abundant than LAP2α in murine heart, independently of the sex. Moreover, we observed a reduction in all LAP2β-like proteins in males compared to females, which was statistically significant for LAP2δ and LAP2γ (Figure 2A,B). In conclusion, the analysis of protein extracts from murine LV suggests a sex-based difference in LAP2 expression.

### 3.4. Dermal Fibroblasts from Patient 1 Harboring the TMPO Variant p.(Gly395Glufs*11), Present a Reduced LAP2α Expression Level Associated with a Decrease in Cell Proliferation

The only cells available from the patients were dermal fibroblasts from patient 1. These cells were maintained in primary culture and compared to dermal fibroblasts from a male control (similar age and cell passage number). The Western blot analysis revealed that the full-length LAP2α protein was significantly decreased (up to 50%) in the patient’s primary cells in comparison to the control fibroblasts (Figure 3A,B). When cells were treated with the proteasome inhibitor MG132, an additional faint band was observed migrating at around 50 kDa, as expected for a highly unstable truncated form of LAP2α induced by the mutation p.(Gly395Glufs*11) (Figure 3C). In situ labeling with antibodies directed against LAP2α allowed the detection of LAP2α in the nucleoplasm of both populations of cells (Figure 3D) but with a significantly weaker signal (35 to 60%) in cells of the patient than in control cells (Figure 3E). The expression of LAP2α was also evaluated at the transcriptional level. We observed a decreased amount (30 to 50%) of mRNA coding for LAP2α in patient’s cells when compared to control cells (Figure 3F). Moreover, a significant increase in LAP2α mRNA level upon the treatment of cells with cycloheximide, that interferes with NMD, suggests degradation of the mutant *TMPO* transcript via nonsense-mediated mRNA decay (NMD) (Figure 3F). To assess whether such a decrease in LAP2α, both at the transcriptional and protein levels, could modify the proliferation capacities of the cells, fibroblasts were immunostained with antibodies directed against the cell proliferation marker Ki67. Indeed, we observed a reduced percentage of Ki67-positive cells in cells from the patient (Figure 3G), suggesting decreased proliferation capacity. This proliferation defect was also confirmed when analyzing the cell growth curves at early, intermediate and late passages (Figure 3H).

### 3.5. Overexpression of TMPO/LAP2 Variants Triggers Abnormal Distinct Cell Phenotypes

As mentioned above, our access to cells from patients harboring *TMPO* mutations was restricted to dermal fibroblasts from patient 1. Therefore, to further explore the pathogenic potential of the three *TMPO* variants, we decided to overexpress them in cell lines. Given the very limited information available on the impact of variants in *TMPO* on different tissues, we did not restrict our cellular study to cardiogenic cells (H9C2) but extended it to skeletal myoblasts (C2C12) and embryonic kidney cells (HEK293). LAP2 proteins (WT versus mutant) were expressed as tagged with DDK. We focused our study on the impact of the *TMPO* mutations on the expression level and some key properties of LAP2s in cells.

#### 3.5.1. Ectopic Expression of the TMPO Indel Variant p.(Gly395Glufs*11) Leads to Full Length LAP2α Quantitative Defects and to Mislocalization of a Truncated LAP2α Protein in Cells

In C2C12 cells, H9C2 cells and HEK293 cells, Western blot protein analysis of whole cell extracts revealed that the WT LAP2α-DDK had an apparent molecular weight of 90 kDa, i.e., slightly higher that the endogenous LAP2α (Figure 4A). A mutant LAP2α protein was detected at a molecular weight of 50 kDa, in accord with the premature stop codon resulting from one nucleotide deletion within the gene sequence (Figure 4A). After the treatment of cells with a hypotonic buffer (0–50 mM NaCl) containing Triton, the mutant LAP2α was largely solubilized together with cytoplasmic components (as GAPDH) instead of WT LAP2α that required higher salt concentrations (150 mM) (Appendix A). Thus, the truncated LAP2α protein showed a reduced nuclear anchorage when compared to WT-LAP2α. We noticed that the expression level of the truncated LAP2α was reduced in comparison to full-length-LAP2α (Figure 4A). Additionally, cell treatment with the proteasome inhibitor MG132 had a positive effect on the expression level of LAP2α p.(Gly395Glufs*11) (Figure 4A, middle and right panels), suggesting some degree of post-translational regulation. We then investigated whether a low expression of the truncated protein might also result from NMD, as previously deduced in dermal fibroblasts from the patient. Therefore, we quantified (by qRT-PCR) the mRNAs encoding specifically the human exogenous LAP2α in rat H9C2 cells, and found a two-fold decrease in the LAP2α-(Gly395Glufs*11) mRNA versus LAP2α-WT-DDK mRNA (Figure 4B). Following cycloheximide treatment, LAP2α p.(Gly395Glufs*11) protein became undetectable (Figure 4B left panel), whereas the LAP2α mutant mRNA level was significantly (>1.6 fold) increased (Figure 4B right panel). Moreover, when we transfected cells with a plasmid allowing to escape the NMD process, a high expression of a truncated LAP2α p.(Gly395Glufs*11) DDK protein (55 kDa) was revealed on immunoblots of whole H9C2 and HEK293 extracts (Figure 4C). This result confirms the role of NMD in reducing mutant LAP2α protein levels.

By immunofluorescence, endogenous and ectopic LAP2α WT were normally detected within the nucleoplasm (Figure 5 A,B). Strikingly, pools of truncated LAP2α, untagged or tagged with DDK (p.G395Efs*-DDK), were detected within the cytoplasm of HEK293, C2C12 or H9C2 cells (Figure 5 A,B). The extra-nuclear localization of LAP2α is consistent with the higher solubilization properties of the variant when compared to the full-length protein (Appendix A). However, the exogenous mutant protein had no impact on the localization of other pathology-associated nuclear envelope proteins. Indeed, Lamin A/C and emerin seemed to localize similarly in cells overexpressing LAP2α WT or mutant (Figure 5B).

In conclusion, both analyses performed in dermal fibroblasts from patient 1 and in transfected cell lines reveal the effect of the frameshift variant g.98927219del_c.1184del_p.(Gly395Glufs*11) to decrease the steady state level of both LAP2α mRNA and protein, due to the NMD process and the proteasome activity, respectively.

#### 3.5.2. Ectopic Expression of the Two Rare Missense TMPO Variants Triggers Nuclear Defects Involving Chromatin-Associated Protein Partners

##### Nuclei Integrity and/or Association with HMGN5 or BAF Are Altered upon the Expression of LAP2 Variants

The expression of the three variants LAP2α p.(Leu124Phe), p.(Ala240Thr) and p.(Arg690Cys) tagged with DDK was efficient in HEK293 cells, although with a slight (but non-significant) reduction when compared to expression of LAP2α WT-DDK (Figure 6A,B). Immunofluorescence showed that these three LAP2α variants localize in the nucleoplasm, as WT LAP2α does (Figure 6C). Unexpectedly, the previously identified variant LAP2α p.(Arg690Cys) specifically induced 2.4-fold more micronuclei (*p* < 0.05) in comparison to WT LAP2α; these micronuclei were labeled for lamin A/C and lamin B1 (Figure 6D,E).

Given their location outside and upstream of the regions required for LAP2α homodimerization (aa 459 to 693) [34], pRb binding (aa 415 to 615) [35] and Lamin A/C binding (aa 615–693) [11], the p.(Leu124Phe) and p.(Ala240Thr) missense mutations are not expected to directly alter these properties/functions of LAP2α. In contrast, due to its location within the LEM domain, the mutation LAP2α p.(Leu124Phe) might modify binding to BAF. In addition, the missense mutation LAP2α p.(Ala240Thr), located within the C-ter region, might modify binding to HMGN5 [19].

Therefore, we decided to assess LAP2α binding to either HMGN5 or BAF using in situ proximity ligation assays (PLA). First, to test proximity events between ectopic LAP2α and endogenous HMGN5, we transfected cells with plasmids encoding LAP2α tagged with DDK and subsequently used specific antibodies directed against DDK (LAP2α) and HMGN5 (Figure 7A,B). Quantitative analysis revealed that the variants p.(Ala240Thr) and p.(Arg690Cys) significantly reduced the steady-state amount of LAP2α in proximity with HMGN5 in nuclei (Figure 7B). These two variants also had a tendency to decrease the amount of proximity events between lamin A/C and LAP2α, although it was not statistically significant due to inter-experiment high variability (Data not shown). Then, to test proximity events between ectopic LAP2α and either endogenous or ectopic BAF, we transfected cells with plasmids encoding LAP2α tagged with DDK alone or together with a plasmid encoding HA-BAF. PLA was done using specific antibodies directed against DDK (LAP2α) and either BAF or HA (Figure 7C,D). Quantitative analysis revealed a significant reduction in the amount of LAP2α-DDK and BAF proximity events per nucleus in both HEK293 and H9C2 cells upon expression of the variant LAP2α p.(Leu124Phe) compared to the WT (Figure 7D).

Since the mutation p.(Leu124Phe) concerns an amino acid shared by all LAP2 isoforms, we also analyzed some properties of the β isoform (Figure 8A). Immunoblots revealed that LAP2β p.(Leu124Phe) had a slight tendency to be less expressed than its WT counterpart (Figure 8B). The variant p.(Leu124Phe) protein localized at the nuclear envelope like WT LAP2β (Figure 8C,D), as expected for an integral protein of the inner nuclear membrane. However, LAP2β p.(Leu124Phe) favored nuclear envelope alterations when compared to WT LAP2β, as shown by a 1.2 to 1.7-fold increase in the frequency of lamin B-negative microbuds in H9C2 cardiomyocytes (Figure 8D) and of severely dysmorphic nuclei in HEK293 cells (Figure 8C; *p* < 0.05). Such alterations suggest fragilization of the nuclear envelope. In cells transfected with LAP2β-DDK and HA-BAF, we observed a significant reduction in the amount of proximity events between LAP2β p.(Leu124Phe) and BAF per nucleus compared to the WT (Figure 8E). The capacity of LAP2β to associate with BAF was also assessed by co-immunoprecipitation (co-IP; Figure 8F). Consistently with our PLA assays, HA-BAF was found associated with LAP2β but reduced amounts of HA-BAF (<40%) were recovered in the immunoprecipitates formed with LAP2β-L124F when compared to LAP2β WT.

Thus, these results suggest an effect of the variant p.(Leu124Phe), leading to a reduction in the binding to BAF for both membranous (β) and nucleoplasmic (α) LAP2 isoforms.

##### The Variant p.(Leu124Phe) Leads to Loss of LAP2α Capacity to Inhibit Gene Transcription Regulated by E2F1

Together with pRb and lamins A/C, LAP2α contributes to downregulate the activity of the transcriptional factor E2F1, with consequences on cell cycle progression [24] and possibly on oxidative metabolism of differentiated cells [25]. Using an E2F reporter gene assay in HEK293 cells, we found that LAP2α WT and the two alpha-specific missense variants LAP2 (p.(Ala240Thr) and p.(Arg690Cys) had a similar capacity to inhibit E2F activity (Figure 9A), whereas p.(Leu124Phe) failed to induce such inhibition (Figure 9A). Using the same E2F reporter gene assay, overexpression of LAP2β either WT or mutant p.(Leu124Phe) had no significant impact on the E2F activity (Figure 9B). We conclude that the variant p.(Leu124Phe) LAP2α fails to inhibit properly the E2F1-regulated gene transcription.

## 4. Discussion

### 4.1. Combined Phenotyping and Gene Panel Sequencing Approaches Support an Association of TMPO with Cardiomyopathies

After a period of rapid progress in the identification of major cardiomyopathy genes based on the study of large informative families and/or homogeneous series, we are facing the challenge of understanding the potential causative role of the rarest genetic causes of this frequent disease.

Whether *TMPO* is or is not a cardiomyopathy-causing gene was unclear. Since the only variant reported so far in two DCM siblings, namely LAP2α p.(Arg690Cys) [21], is too prevalent in gnomAD, *TMPO* has been subsequently excluded from most cardiomyopathy diagnostic gene panels [5].

In our cardiomyopathy series, three novel heterozygous *TMPO* variants (c.1184del_p.(Gly395Glufs*11), c.370C>T_p.(Leu124Phe) and c.718G>A_p.(Ala240Thr), were identified in 6/5000 patients belonging to five unrelated families, in the absence of other disease-causing variants in the major CM genes [26] and in exome sequencing in some of them. These variants affect LAP2 amino acids highly conserved in mammalians and occur infrequently in the general population (1-7/251418 alleles in gnomAD).

*TMPO* variants with a predicted Loss of Function (pLoF) due to the generation of prematurely truncated proteins are reported in gnomAD database, suggesting that the gene is not a constrained. However, all the *TMPO* pLoF variants in gnomAD show an allele frequency below the threshold (0.01%) used in the context of cardiomyopathy. Furthermore, considering the incomplete penetrance of cardiomyopathy depending on age and sex, some of the gnomAD carriers of pLoF in *TMPO* might be at an asymptomatic stage of the disease. We propose that the severity of HCM associated with the p.(Gly395Glufs*11) variant would rely on (i) *TMPO* haploinsufficiency combined with other genetic co-factors and/or (ii) a potential toxic effect of the truncated protein that we showed to be mislocalized in the cytoplasm.

The fact that the *TMPO* variant p.(Leu124Phe) could be a genetic contributor to DCM, is supported by the absence of additional gene variants identified by exome sequencing in Patient 6. However, since severe phenotype in newborn patients is frequently due to co-dominance of genes, as previously reported both in DCM and HCM [36,37,38], we suggest that the extreme severity and precocity of the phenotype for patient 5 (death at 2.5 months) may be due to a cumulative effect of probably genetic and/or non-genetic factors yet unidentified to date despite exome sequencing.

Finally, the fact that mutations of a single gene lead to distinct types of cardiomyopathies is not restricted to *TMPO* and has been reported for most of the other cardiomyopathy genes [26].

### 4.2. Sex influence on Disease Occurrence

It is well known that HCM or DCM have a lower penetrance in women than in men [39,40]. In some cases (as for *EMD* that encodes emerin, [41]), the distinct penetrance is attributed to the localization of the causative gene on the X chromosome. Alternatively, although not causative of the disease, a gene located on one chromosome X presenting specific single nucleotide polymorphism (SNP) or epigenetic marks might contribute to modulating the severity of the disease [42]. Sex hormones influence the prevalence and severity of cardiac disease, due to their involvement in regulating diverse metabolic pathways [42]. Interestingly, a comparative analysis of the transcriptome expression profile done on left ventricles from a healthy group (26 men and 17 women, post-mortem or organ donors aged between 21 to 68 years) revealed sexually dimorphic genes, including genes located on sex chromosomes but also autosomal genes [43].

*TMPO* is an autosomal gene, located on chromosome 12 (cytogenetic band12q23.1) in humans. In our cohort, the mothers of the patients (2, 3, 4 and 5) who transmitted the *TMPO* mutations to their sons showed no phenotype to date. Accordingly, the mouse animal model of *TMPO*/LAP2α knock-out revealed systolic dysfunction exclusively in male mice [20]. We hypothesized that LAP2 protein expression could be differentially modulated in the hearts of males vs. females, which could increase the occurrence and/or the severity of the cardiomyopathies associated with *TMPO*/LAP2α variants specifically in males. Our data from heart analysis in young adult mice support this hypothesis, since the proteins LAP2α, δ and γ were significantly less abundant in the left ventricle of males vs. females. Our results will have to be expanded by performing similar analyses in mice of different ages, considering different sub-compartments of the heart (atria, septum, etc.), and possibly extended to LAP2 partners and other nuclear envelope-associated proteins, followed by validation in human tissues.

### 4.3. Cellular Studies Reveal Pathogenic Mechanisms Induced by Three Novel Rare TMPO Variants

LAP2 functions are indeed complex with tissue specificity and not fully understood [18,44]. Taylor et al. who described *TMPO* p.(Arg690Cys) assumed this variant could cause DCM based on the evaluation of only one function (interaction of LAP2α p.(Arg690Cys) with the lamin A/C terminus) [21], although it is now known that this is a frequent and functional polymorphism in Latin America, and may be considered as a modifier gene [45,46].

The three *TMPO* variants reported here affect LAP2 amino acids highly conserved in mammalians and occur infrequently in the general population (1-7/251418 alleles in gnomAD). We found that the three rare *TMPO* variants p.(Gly395Glufs*11), p.(Leu124Phe), p.(Ala240Thr) and the frequent variant p.(Arg690Cys) induced abnormal and distinct cellular phenotypes. Thus, our data validate the in silico predictions of pathogenicity for the new *TMPO* variants and also illustrate the specificity of the different domains of LAP2 proteins (LEM domain versus C-ter region).

#### 4.3.1. The p.(Gly395Glufs*11) LAP2α Variant Induces Haploinsufficiency

The main consequence of the p.(Gly395Glufs*11) variant is LAP2α haploinsufficiency which is associated with decreased cell proliferation of dermal fibroblasts. A similar effect was previously reported by downregulating the LAP2 protein level using siRNA [47]. LAP2α haploinsufficiency is mainly due to transcriptional regulation (NMD), although the degradation of the protein by the proteasome might also contribute. Interestingly, NMD has been reported as a major mechanism in HCM associated with cardiac myosin-binding protein C (*MYBPC3*) mutations [48]. Lower levels of LAP2α might impact all the functions normally regulated by LAP2α together with its partners, including cell cycle regulation, genome organization and gene expression [18,24,49]. In addition, a truncated LAP2α protein was observed in our cellular models upon expression of the p.(Gly395Glufs*11) variant. Even after treatment with the proteasome inhibitor MG132, only a faint signal compatible with a potential truncated LAP2α protein was detected in primary human dermal fibroblasts from Patient 1. Nevertheless, one cannot exclude that truncated LAP2α protein would accumulate more specifically in the heart than in dermal fibroblasts, for instance upon deregulation of the NMD process and/or the proteasome activity consequently to particular environmental conditions (mechanical/physical stress, etc.). Interestingly, the downregulation of proteasome activity has been reported during heart ageing in the rat animal model [50]. Whether this could promote the accumulation of truncated LAP2α protein over time in the heart in humans should be determined. Our data showed that in that case, truncated LAP2α would mislocalize to the cytoplasmic compartment with a potential toxic effect. Altogether, the cellular defects triggered by p.(Gly395Glufs*11) variant may contribute to myocyte hypertrophy and disarray which are typical of HCM [51], and represent a potential future line of investigation.

Intriguingly, another *TMPO* variant (a single base pair insertion) was predicted to cause a frameshift and premature stop codon after the residue Thr99, was recently reported as associated with another pathology, nonalcoholic fatty liver disease (NAFLD) [52]. Although this variant was identified in two monozygotic twins, only one of them developed the disease, suggesting the p.(Thr99fs) variant as being not a monogenic cause but a predisposition factor to NAFLD [52]. Interestingly, while the variant p.(Thr99fs) is expected to impact all LAP2 isoforms, the variant p.(Gly395Glufs*11) concerns the isoform LAP2α, a difference which might partly explain the different impact of these two variants in humans. Of note, the fact that different mutations in a gene associated with diseases targeting different tissues is not new and has been particularly illustrated with the *LMNA* gene, encoding the LAP2α partners lamin A/C [53].

#### 4.3.2. The LAP2alpha-Specific Variants p.(Ala240Thr) and p.(Arg690Cys) Modify LAP2α Association with the Genome Wide Organization Regulator, HMGN5

In accordance with their localization in the C-ter region of LAP2α, the variants p.(Ala240Thr) and p.(Arg690Cys) induced a reduced amount of proximity events between LAP2α and HMGN5 in cells. It is known that, by binding both the nucleosomes and LAP2α, HMGN5 regulates the genome-wide organization [19] and the capacity of the nucleus to withstand mechanical forces, as occurring in the contracting heart [23]. Thus, in the presence of LAP2α p.(Ala240Thr) or p.(Arg690Cys), a consequent alteration of the role of HMGN5 might contribute to abnormal properties of cardiomyocytes. However, for the variant p.(Arg690Cys) LAP2α, its high frequency in the GnomAD population indicates that its potential influence in cardiac tissue would depend on additional external/environmental conditions. In the literature, this polymorphic LAP2α variant was reported to increase auto-immune disease susceptibility (neuromyelitis optica spectrum disorder) in an admixed Mexican population [46]. It was also shown to impact the nuclear circularity of leukocytes and to modulate the abnormal nuclear phenotypes triggered by a lamin A/C truncating mutation causing DCM [45]. In our study, we observed that the variant p.(Arg690Cys) LAP2α frequently induced micronuclei containing lamins A/C and B1, which might reflect a mitotic defect and contribute to genomic instability [54]. Since these micronuclei were observed exclusively in embryonic cells (HEK293), we suggest that p.(Arg690Cys) LAP2α might have an impact in organs at specific stages of active proliferation rather than a universal pathogenic effect.

#### 4.3.3. The LEM-Domain Variant p.(Leu124Phe) Modifies LAP2α-Mediated Regulation of BAF and E2F1

The missense variant p.(Leu124Phe) has mechanistic consequences on both LAP2 major isoforms, α and β. With LAP2α, we have shown that p.(Leu124Phe) induced an altered E2F-regulated gene transcription inhibition. This defect is associated with a reduced amount of proximity events between LAP2α mutant and BAF, possibly due to altered 3D packing of the two LEM alpha helices, which in turn, could modify the association with lamin A/C [27]. E2F deregulation is expected to impact the regulation of the cell cycle as well as cardiac oxidative metabolism [24,25]. For the other isoform, LAP2β, both proximity ligation assays and co-immunoprecipitation experiments supported a negative impact of the variant p.(Leu124Phe) on binding to BAF. In addition, this LAP2β variant specifically increased the frequency of dysmorphic nuclei. Similarly, the p.(Leu13Arg) variant of LEMD2, another membranous protein with a LEM domain, was reported to induce an increased frequency of nuclear membrane invaginations in fibroblasts and heart tissue from patients diagnosed with arrhythmic cardiomyopathy [55]. Of note, the leucine residues mutated in LAP2 (p.(Leu124)) and in LEMD2 (p.(Leu13)) are located at the same highly evolutionary conserved position within the LEM domain. These data highlight the key role of the LEM domain for the two inner nuclear membrane proteins, LAP2 and LEMD2, to properly regulate the NE structure. In the presence of the LEM variant, the alteration of the NE structure might be sustained by a change in the topology of NE-associated proteins such as lamin B1 or HDAC3. HDAC3 together with lamins contributes to tether the peripheral heterochromatin and has been proposed to prevent precocious cardiac progenitor differentiation [56]. Thus, it will be interesting to assess the impact of p.(Leu124Phe) LAP2β on the expression of NE-associated proteins, chromatin attachment at the NE and/or gene silencing.

### 4.4. Relevance of the LEM Domain of Nuclear Proteins in Cardiac Pathophysiology

It is well established that cardiomyopathies can be caused by mutated nuclear envelope-associated proteins such as lamins A/C [8], the inner nuclear membrane LAP1 (lamina-associated polypeptide 1) [57] and LEM domain proteins. Among the latter, LEMD2 and emerin are inserted within the inner nuclear membrane, like LAP2β, but interact with lamins A/C like LAP2α [15,58]. The LEMD2 recessive mutation p.(Leu13Arg) is associated with juvenile cataract and a severe form of arrhythmic cardiomyopathy [55]. While most emerin mutations lead to haploinsufficiency and a combined skeletal and cardiac muscle phenotype (X-EDMD, Emery Dreifuss muscular dystrophy [41]), three mutations in the LEM domain of emerin were identified in patients presenting exclusively atrial cardiac defects [59,60,61]. These include the p.(Lys37del) mutation, concerning an amino acid conserved in LAP2, MAN1 and LEMD2. This mutation leads to incorrect folding of the emerin LEM domain and weakened affinity to bind BAF in vitro [61]. Here, we report the first mutation in the LEM domain of LAP2 that was predicted to destabilize the folding of the LAP2 N-ter region. The LEM mutation affects both the nucleoplasmic isoform α and the membranous isoform β. Interestingly, the Lysine 37 of emerin and the Leucine 124 of LAP2 face each other at the 3D surface of two large parallel α-helices [16]. In accordance with the impact of p.(Lys37del) emerin mutation [61], we show that the LAP2 p.(Leu124Phe) variant reduces the proximity and/or association of LAP2 with BAF in cells. Our data suggest that the LEM domain might be important in the physiology and pathophysiology of the heart, revealing putative common pathophysiological mechanisms associated with mutations in LEM-domain proteins.

### 4.5. Study Limitations

The findings of this study have to be seen in the light of some limitations.

The first concerns the family pedigrees of the six patients with cardiomyopathy associated with *TMPO*/LAP2 variants. Data are limited to those involving only two generations, patients with siblings and parents, for whom clinical and genetic information is sometimes missing. Additional efforts should be made to explain to patients the potential benefits to their relatives of conducting clinical and genetic follow up. Expanded family pedigrees would allow a more accurate assessment of the mode of inheritance and the role of *TMPO* mutations associated with cardiomyopathies.

The second limitation of our study concerns the limited number of patients (six), which does not allow us to generalize our results. In particular, although our six patients were men, it cannot be stated that cardiomyopathies related to *TMPO* mutations will occur exclusively in men. The systematic inclusion of *TMPO* in the cardiomyopathy-targeted gene panels should help us to clarify this point.

Thirdly, although exome sequencing has not revealed variants in genes other than *TMPO* which could explain the cardiomyopathy phenotype in patients, one must keep in mind that the overall diagnostic yield of this technique is typically limited even in familial cardiomyopathies. We cannot exclude the presence of a pathogenic variant in non-sequenced regions such as introns, regulatory or intergenic regions, and therefore, not finding other candidate variants does not rule out the possibility of other causal mutations.

Finally, we must acknowledge the limitation of our choice to evaluate in cells only a few functions of LAP2α according to the functional domain affected by a given mutation. Indeed, the role of LAP2α is complex; it contributes to the regulation of many functions such as chromatin organization/dynamics, gene expression and cell proliferation, and in a tissue or cellular context-dependent manner [9,18,19,20,24,35,44,47,62]. Further exploration of the impact of *TMPO*/LAP2 variants in different tissues or in cells of different lineages will help to clarify the role of LAP2 proteins.

## 5. Conclusions

Our study in cells supports a pathogenic role for the three new rare *TMPO* variants p.(Gly395Glufs*11), p.(Ala240Thr) and p.(Leu124Phe) identified in five cardiomyopathy families, in particular by altering cell proliferation and/or proper function of chromatin-associated proteins such as BAF, HMGN5 and E2F1, with expected consequences on gene structure/expression.

Altogether, our data suggest that *TMPO* variants might be associated with human diseases, namely cardiomyopathies, which appears as an exciting line of future investigation. Finally, our data suggest sex-based differences in LAP2 expression, which might contribute to modulating cardiomyopathy phenotype expression in humans, in accordance with the mice model of *TMPO* deficiency [20]).

## Figures and Tables

**Figure 1 cells-12-00337-f001:**
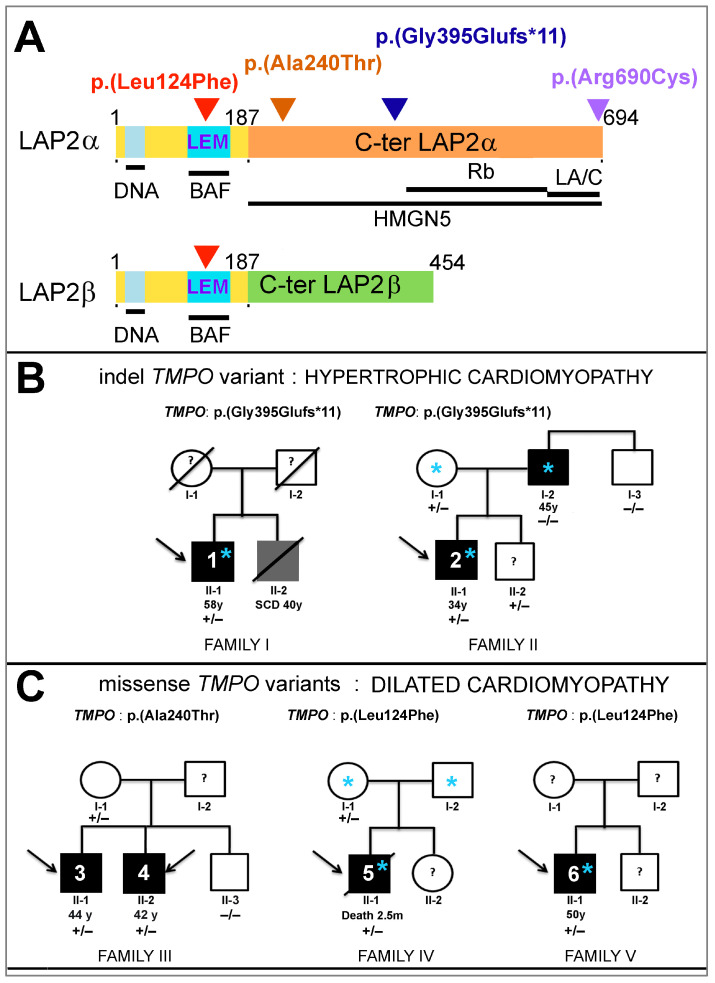
Identification of three rare *TMPO*/LAP2 variants in patients with cardiomyopathies. (**A**) Localization of variants in LAP2 proteins. LAP2α and LAP2β have 694 and 454 amino acids (a.a), respectively. The region common to all LAP2 isoforms (α, β…) includes the N-ter (a.a 1–187) encoded by exons 1, 2 and 3 (in yellow) and contains a LEM-like (in pale blue) and a LEM domain (in blue). The C-ter regions specific to LAP2α (exon 4) and LAP2β (exons 5–10) are depicted in orange and green, respectively. The mutations identified in patients with cardiomyopathies and their position on the proteins are indicated (arrow heads). The LAP2α domains involved in binding DNA or the protein partners BAF, Rb, lamins A/C (LA/C) and HMGN5 are underlined. (**B**,**C**) Pedigrees of the five families carrying new rare *TMPO* mutations. Generations are identified by their Roman numerals below the symbol. Arabic numerals denote each individual in a generation. Squares represent men, and circles represent women. Index cases were indicated with an arrow. Open and filled symbols are unaffected and affected individuals, respectively; diagonal lines indicate deceased subjects. Age (years, y; months, m) at diagnosis or death is indicated, SCD: sudden cardiac death. Echocardiography and electrocardiograms were reported as normal for the carrier mothers in Families II, III and IV. Stars * (in blue) indicate patients that underwent exome sequencing. + means an allele carrying the LAP2 mutation of interest, − means no LAP2 mutation.

**Figure 2 cells-12-00337-f002:**
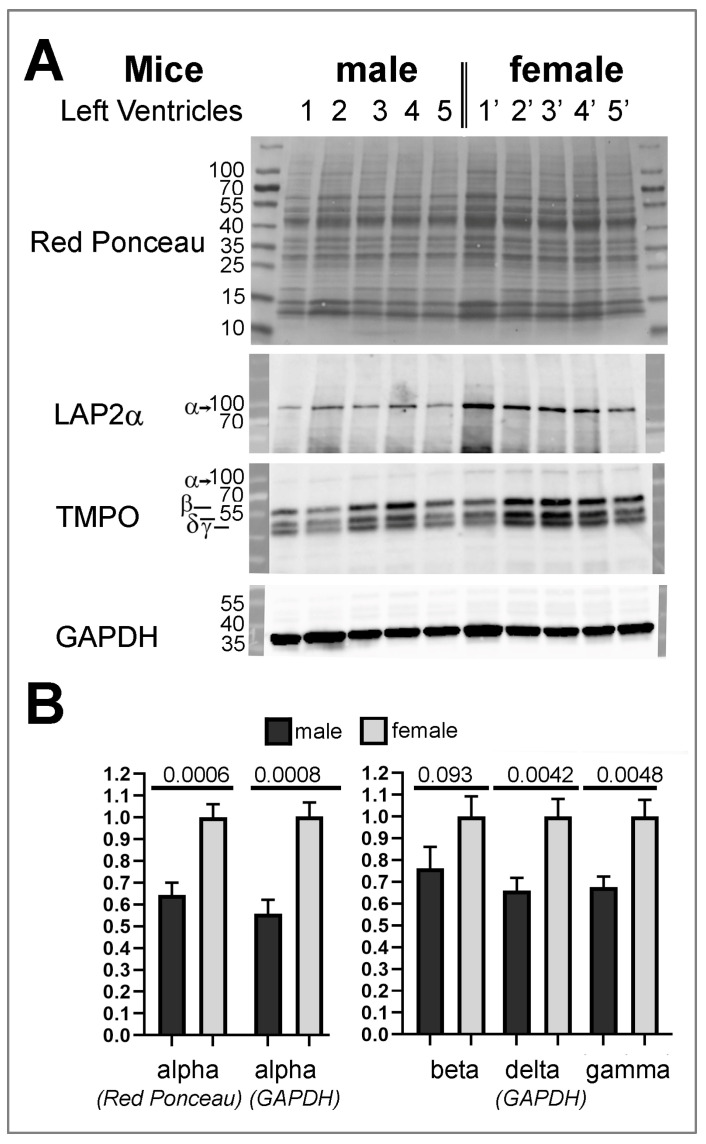
Relative expression level of LAP2α and LAP2β, δ, γ proteins in the hearts of male vs. female mice aged of 9 weeks. (**A**), Western blots are shown on whole extracts of LV from 5 male mice (1–5) and 5 female mice (1′–5′) stained with red ponceau and incubated first with anti-LAP2α antibodies and subsequently with *TMPO* antibodies (that recognize all LAP2 proteins). Red Ponceau staining or GAPDH were used to normalize ECL signals (Additional results related to 5 other male mice (6–10) and 5 other female mice (6′–10′) are presented in Appendix A). (**B**) The graph shows ECL signal quantification related to A) and Appendix A (mean ± s.e.m; n = 10). For statistical analyses, precise *p*-values are indicated.

**Figure 3 cells-12-00337-f003:**
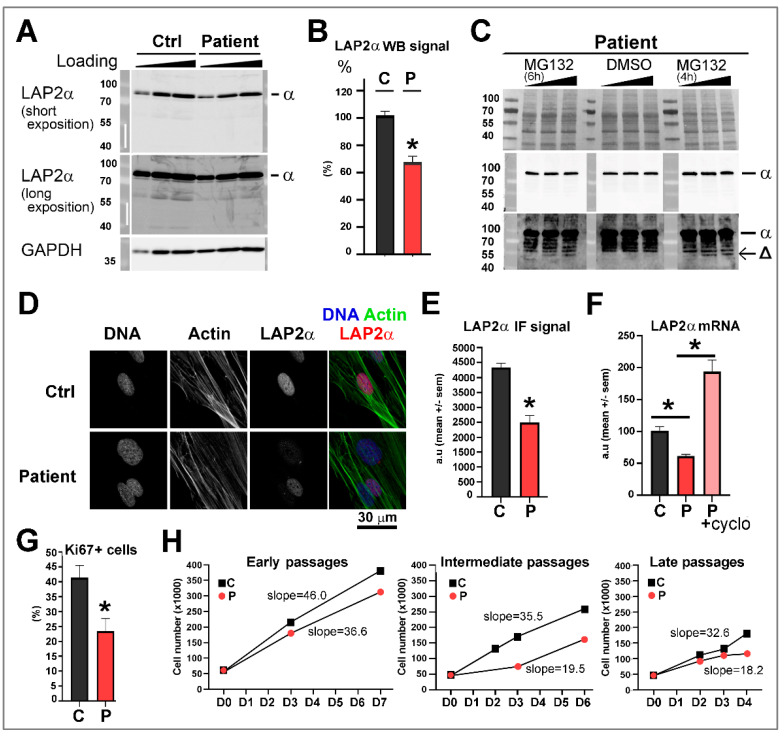
Expression of LAP2α in dermal fibroblasts of a patient carrying the *TMPO* mutation p.(Gly395Glufs*11). (**A**) Whole-cell extracts of fibroblasts from one healthy male control (Ctrl) and patient 1 were analyzed by Western blot using anti-LAP2α antibodies and anti-GAPDH antibodies to control loading of samples. (**B**) The graph shows ECL signal quantification (mean ± s.e.m.) of full-length LAP2α proteins expressed in human dermal fibroblasts (n = 8 experiments, GAPDH or Actin used as reference proteins). * *p* < 0.05. (**C**) Whole-cell extracts of fibroblasts from patient 1 pretreated with or without 20 µM MG132 for either 4h or 6h, as indicated, and analyzed by Western blot using anti-LAP2α antibodies. Are shown two expositions for the ECL revelation (short exposition in the middle panel; longer exposition in the lower panel). Signs α and Δ indicate the detection of full-length LAP2α and truncated LAP2α, respectively. Red ponceau is also shown in the upper panel. (**D**) Dermal fibroblasts were labeled with anti-LAP2α antibodies (red) and phalloidin to reveal the actin network (green). DNA was stained with Hoechst (blue). Scale bar, 30 µm as indicated. (**E**) Relative intensity of LAP2α immunofluorescence (IF) signals per nucleus in fibroblasts from the control (C) and the patient (P) (n = 5 experiments, n = 120–170 nuclei for exps 1–3, n = 520–650 nuclei for exps 4-5). * *p* < 0.05 (Mann–Whitney test). (**F**) LAP2α mRNA quantification in fibroblasts from one healthy male control (Ctrl) and patient 1, pre-incubated with (+) or without (−) cycloheximide (n = 4 experiments). HPRT was used as a reference gene, as validated with the NormFinder algorithm. * *p* < 0.05 (Kruskal–Walllis test). (**G**) Percentage of cells (control (C; passages 8–12) versus patient (P; passages 6–13)) positively stained upon immunofluorescence performed with the antibodies anti-Ki67 (n = 5 experiments, n = 170–300 nuclei for exps 1–4; n = 430–630 nuclei for exp5) * *p* < 0.05 (Paired t-test). (**H**) Growth curves of dermal fibroblasts. Cells were seeded at day 0 (D0) at 50.000 cells/well at early (p4 (patient) and p9 (Ctrl)), intermediate (p15 (patient) and p14 (Ctrl)) or late passages (p18 (patient) and p19 (Ctrl)). The graphs show the number of cells counted at 2, 3, 4 and/or 7 days post-seeding, as indicated. Values for the slope of the growth curves are indicated.

**Figure 4 cells-12-00337-f004:**
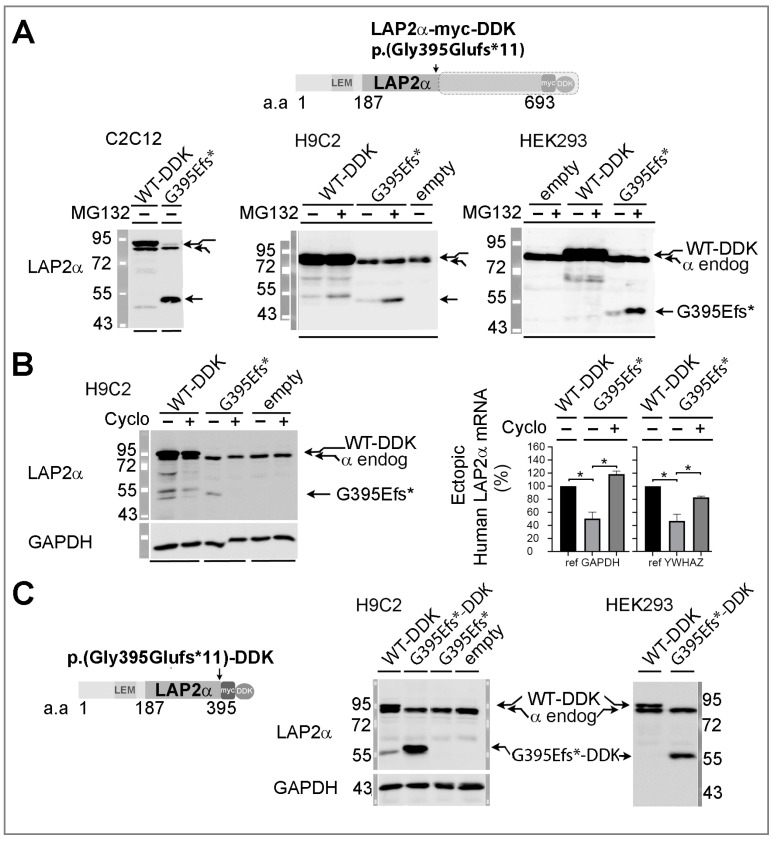
Expression levels of full-length and truncated LAP2α in cells. (**A**) Schematic secondary structure of exogenous p.(Gly395Glufs*11) (G395Efs*) LAP2α (Top panel). LAP2α Western blot on whole cell extracts of C2C12 cells, H9C2 cells or HEK293 cells overexpressing either wild-type (WT-DDK) or mutant (G395Efs*) LAP2α or transfected with an empty vector (empty). Cells were pre-incubated with (+) or without (−) 20 µM MG132 for 3h, as indicated. (**B**) H9C2 rat cells overexpressing WT or G395Efs* LAP2α were pre-incubated (+) or not (−) with 100 µg/mL cycloheximide (Cyclo) for 6h; Western blot (left) and ectopic LAP2α mRNA quantification (right). The graphs show the mRNA amount (relative to WT; mean ± s.e.m.) of ectopic human LAP2α (n = 6 experiments); GAPDH or YWHAZ was used as a reference gene, as indicated. * *p* < 0.05. (**C**) Schematic secondary structure of exogenous p.(Gly395Glufs*11-DDK) (G395Efs*-DDK) LAP2α proteins (Left panel); for suppressing NMD, the original mutated sequence around G395 was kept and followed immediately by the DDK tag. LAP2α Western blot on whole extracts of H9C2 cells or HEK293 cells overexpressing WT-DDK, G395Efs*-DDK or G395Efs* LAP2α proteins; with GAPDH used as a reference protein.

**Figure 5 cells-12-00337-f005:**
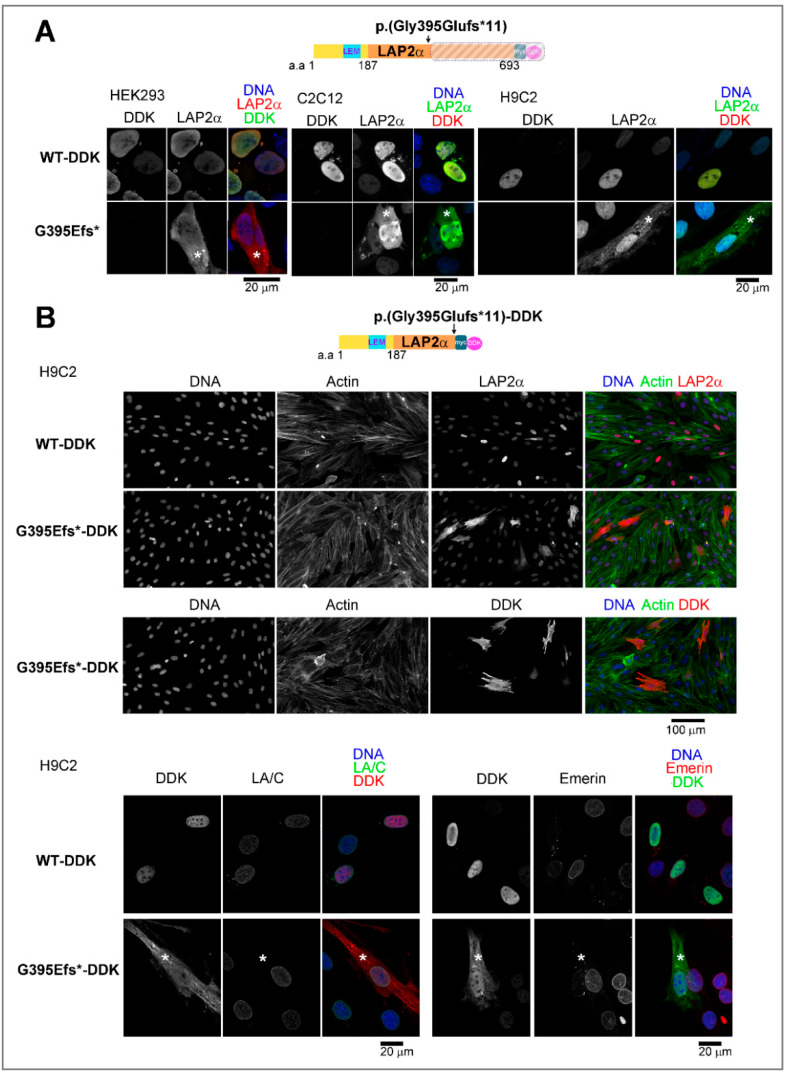
Abnormal cyto-nucleoplasmic distribution of the truncated LAP2α variant. (**A**) HEK293, C2C12 and H9C2 cells overexpressing wild-type (WT-DDK) or mutant p.(Gly395Glufs*11) (G395E*) LAP2α labeled with anti-DDK and anti-LAP2α antibodies, as indicated. (**B**) H9C2 cells overexpressing WT-DDK or mutant G395E*-DDK LAP2α were labeled with different mixtures of antibodies, anti-LAP2α or anti-DDK antibodies (red) and phalloidin (Actin, green) (Upper panels), anti-DDK (red) plus anti-lamin A/C (LA/C, green) antibodies (lower panels, left), or anti-DDK (green) plus anti-emerin (red) antibodies (lower panels, right), as indicated. (**A**,**B**) DNA was stained with Hoechst (blue). Asterisks show cytoplasmic localization of the truncated LAP2α protein. Scale bars = 20 or 100 µm, as indicated.

**Figure 6 cells-12-00337-f006:**
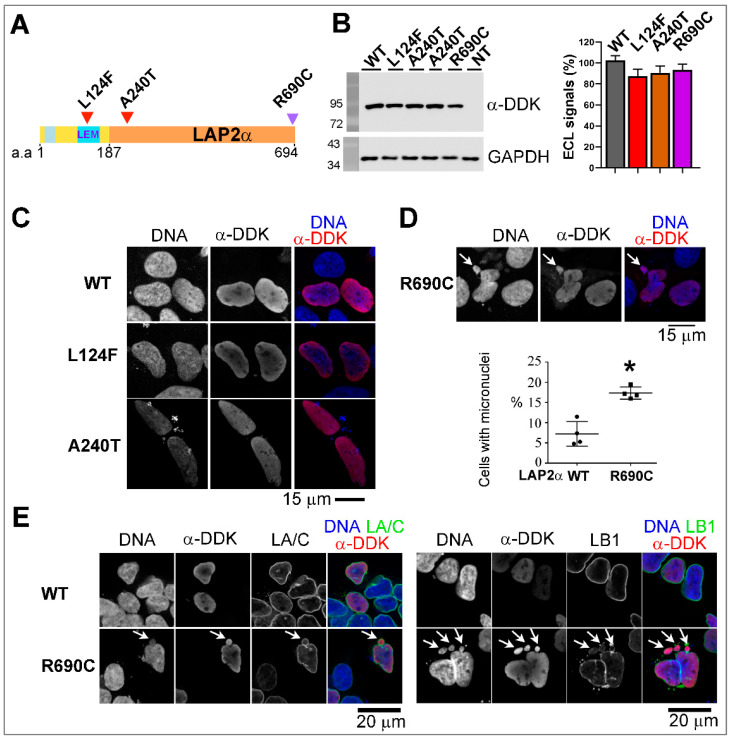
Impact of LAP2α missense variants expression on nuclear phenotypes. (**A**) Secondary structure of exogenous LAP2α. Arrowheads indicate the position of the mutations p.(Leu124Phe) (L124F), p.(Ala240Thr) (A240T) and p.(Arg690Cys) (R690C). (**B**) Western blot on whole extracts of HEK293 cells either non-transfected (NT) or overexpressing wild-type (WT) or mutant LAP2α-DDK proteins using anti-DDK or anti-GAPDH antibodies. The graph shows ECL signal quantification (relative to DDK; mean ± s.e.m; n = 9 (WT, R690C), n = 8 (L124F) or n = 5 (A240T) experiments. (**C**) HEK293 cells overexpressing WT or mutant (L124F, A240T) LAP2α-DDK labeled with anti-DDK (red) antibodies. (**D**) HEK293 cells overexpressing mutant (R690C) LAP2α-DDK labeled with anti-DDK (red) antibodies. Arrows show micronuclei. The graph shows the percentage of cells with micronuclei (mean ± SD, n = 4 experiments). * *p* < 0.05 (Mann–Whitney test). (**E**) HEK293 cells overexpressing WT or mutant (R690C) LAP2α-DDK labeled with anti-DDK (red) antibodies mixed with anti-lamin A/C (LA/C, green) or anti-lamin B1 (LB1, green) antibodies. Arrows show micronuclei. In (**C**–**E**), DNA was stained with Hoechst (blue). Scale bars = 15 or 20 µm, as indicated.

**Figure 7 cells-12-00337-f007:**
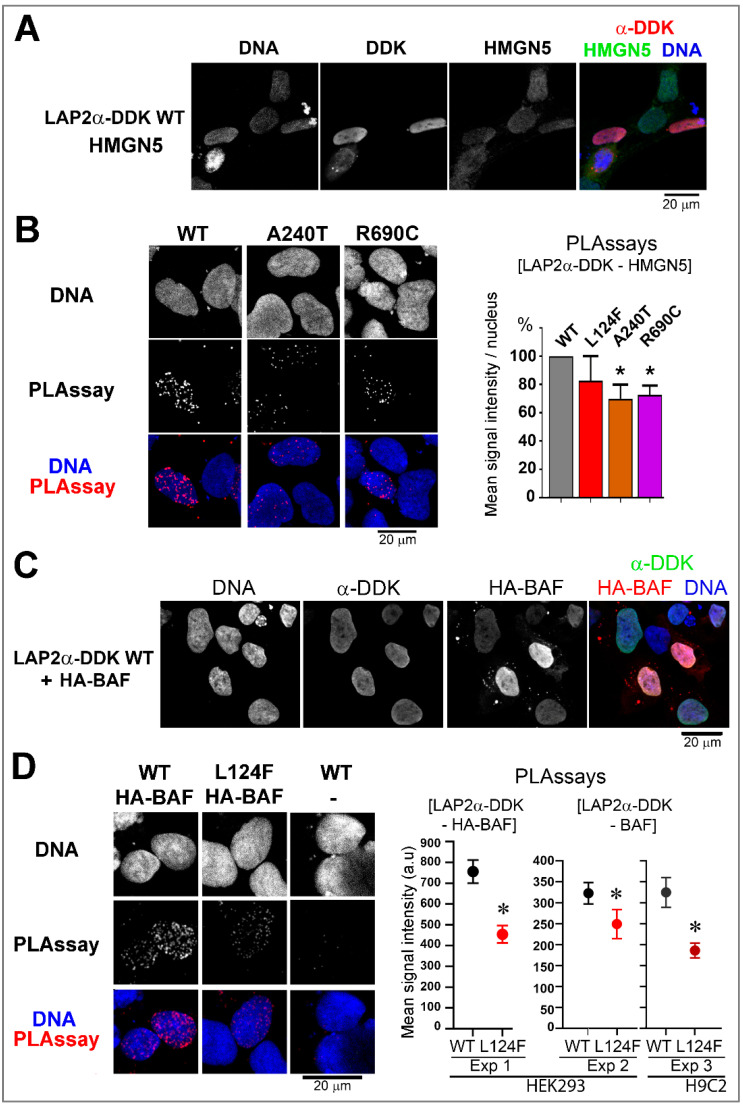
Impact of LAP2α missense variants on in situ proximity events between LAP2α and HMGN5 or BAF. (**A**) Immunofluorescence of HEK293 cells expressing WT LAP2α DDK were labeled with anti-DDK (red) and anti-HMGN5 (green) antibodies. (**B**) HEK293 cells expressing WT or mutant (A240T, R690C) LAP2α-DDK were processed for PLA to reveal [LAP2α-DDK—HMGN5] proximity events (red dots). The graph represents the relative mean of PLA signal intensity/nucleus (mean ± s.e.m.; n = 3 (R690C) or n = 4 (WT, L124F, A240T) experiments). * *p* < 0.05 (Mann–Whitney test). (**C**)**,** Immunofluorescence of HEK293 cells co-expressing WT LAP2α-DDK and HA-BAF were labeled with anti-DDK (green) and anti-HA (red) antibodies. (**D**) HEK293 cells co-expressing WT or mutant (L124F) LAP2α-DDK and HA-BAF were processed for PLA to reveal [LAP2α-DDK—HA-BAF] proximity events (red dots). The graph (left panel, Exp1) represents the mean of PLA signal intensity/nucleus (mean ± s.e.m., n = 200 to 300 nuclei). Alternatively, cells that expressed WT or mutant (L124F) LAP2α-DDK were processed for PLA to reveal [LAP2α-DDK—endogenous BAF] proximity events (red dots) in HEK293 (Exp 2) and H9C2 (Exp 3) cells. The graphs represent the mean of PLA signal intensity/nucleus related to WT (in black or grey) and mutant (L124F, in red) LAP2α-DDK (mean ± s.e.m., n = 60 to 130 nuclei). * *p* < 0.05 (Mann–Whitney test). In (**A**–**D**) DNA was stained with Hoechst (blue). Scale bars = 20 µm.

**Figure 8 cells-12-00337-f008:**
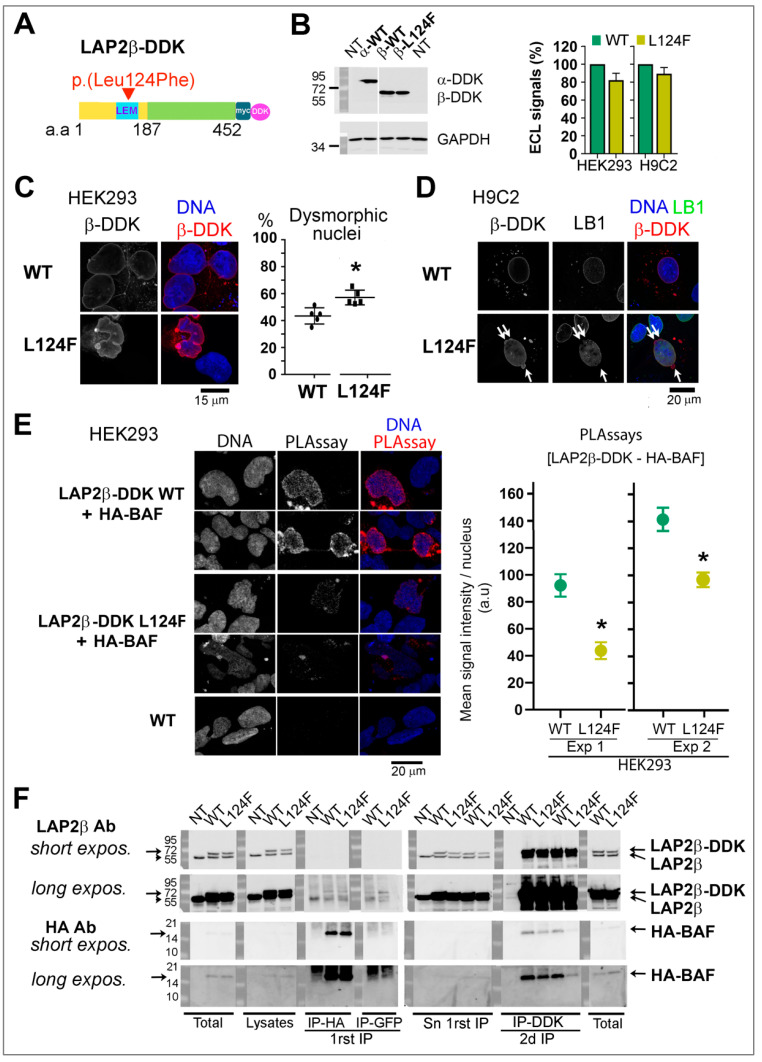
Impact of p.(Leu124Phe) LAP2β on nuclear phenotypes and association with BAF. (**A**) Secondary structure of exogenous LAP2β. Arrow-head indicates the position of p.(Leu124Phe) (L124F). (**B**) Western blot on whole extracts of HEK293 cells either non-transfected (NT) or expressing LAP2α-DDK wild-type (WT) or LAP2β-DDK wild-type (WT) or mutant (L124F) using anti-DDK or anti-GAPDH antibodies. WT and Mutant LAP2β-DDK migrated at ~55kDa. The graph shows ECL signal quantification of LAP2β-DDK relative to DDK (mean± SD of n = 6 or 5 experiments for HEK293 and H9C2 cells, respectively. (**C**) HEK293 cells overexpressing WT or L124F LAP2β-DDK labeled with anti-DDK (red) antibodies. The graph shows the percentage of cells with severe dysmorphic nuclei (mean ± SD, n = 5 experiments). * *p* < 0.05 (Mann–Whitney test). (**D**) H9C2 cells overexpressing WT or L124F LAP2β-DDK labeled with anti-DDK (red) and anti-Lamin B1 (LB1, green) antibodies. Arrows show LB1-negative microbuds. (**E**) HEK293 cells co-expressing WT or mutant (L124F) LAP2β-DDK and HA-BAF were processed for PLA to reveal [LAP2β-DDK—HA-BAF] proximity events (red dots). The graphs (right panel) represent the mean of PLA signal intensity/nucleus related to WT (in green) and mutant (L124F, in yellow) LAP2β-DDK for two experiments (Exp 1, 2) (mean ± s.e.m.; n = 140 to 280 nuclei). * *p* < 0.05 (Mann–Whitney test). In (**C**–**E**) DNA was stained with Hoechst (blue). Scale bars = 15 or 20 µm, as indicated. (**F**) Non transfected HEK293 cells (NT) and cells co-expressing WT or mutant (L124F) LAP2β-DDK together with HA-BAF were processed for two rounds of immunoprecipitation; first with mouse anti-HA antibodies (IP-HA) and second with mouse anti-FLAG M2 antibodies (IP-DDK). Total lysates (Total), soluble lysates (Lysates), and supernatants (Sn) and pellets (IP) recovered after immunoprecipitation were analyzed by immunoblotting. Rabbit anti-LAP2β antibodies (LAP2β Ab) and rabbit anti-HA antibodies (HA Ab) were used to detect LAP2β (endogenous or tagged with DDK) and HA-BAF, respectively. As a negative control, mouse anti-GFP antibodies (IP-GFP) captured neither HA-BAF nor LAP2β-DDK.

**Figure 9 cells-12-00337-f009:**
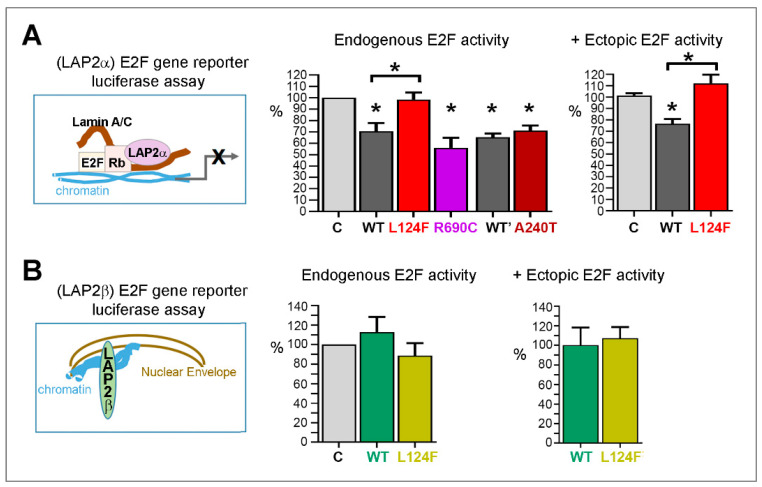
Impact of the p.(Leu124Phe) mutation on the capacity of LAP2s to inhibit E2F activity. Reporter E2F-luciferase activity driven by binding of WT or mutant (L124F, R690C A240T) LAP2α DDK (**A**) and WT or mutant L124F LAP2β-DDK (**B**) to E2F target sequences. LAP2-DDK proteins were expressed in the absence or presence of ectopic HA-E2F1 in HEK293 cells. Results are expressed in % activity versus cells co-expressing E2F-luciferase and an empty pcdna vector (C). N = 3 to 5 independent experiments. * *p* < 0.05 (Mann–Whitney test).

## Data Availability

Not applicable.

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
