# Peer review of "Abnormal Cellular Phenotypes Induced by Three TMPO/LAP2 Variants Identified in Men with Cardiomyopathies"

_cells, 2023, doi:10.3390/cells12020337_

Round 1

Reviewer 1 Report

The authors present an interesting paper addressing the yet unsolved issue of whether TMPO mutations are causal of cardiomyopathy.  They identified 3 different TMPO mutations in 6 patients (DCM and HCM) and present a large amount of experimental evidence of the consequences of 3 different mutations at the molecular and cellular levels. Altogether, their results strongly suggest that TMPO mutations may cause both hypertrophic or dilated cardiomyopathy, and that these mutations show a sex effect, being more severe or perhaps only expressed in males. They make a strong case of causality, however some issues should be addressed before publication.

General issues:

1)     The authors should further enrich the discussion section in the light of the incomplete knowledge of LAP2 functions.  LAP2 functions are indeed complex, with tissue specificity and not fully understood (doi: 10.7554/eLife.63476.). Taylor et al. who described TMPO Arg690Cys assumed this variant could cause DCM based on the evaluation of only one function (interaction of LAP2 690Cys with the Lamin A/C terminus), although it is now known that this is a frequent and functional polymorphism in Latin America, and may be considered a modifier gene (doi: 10.3390/ijms232113626.; doi: 10.3389/fgene.2021.647343. eCollection 2021).  While the authors designed their experiments to assess several functions according to the functional domain affected by the mutation (in the case of missense mutations), they must acknowledge the limitations of assessing these functions under the scenario of incomplete understanding of LAP2.

2)     The authors should interpret their findings considering that the overall diagnostic yield of exome sequencing in cardiomyopathies is relatively low, even in familial DCM or HCM.  Thus, not finding other candidate variants does not rule out the possibility of other causal mutations.

3)     The authors should improve the structure of the manuscript, keeping discussion statements in the discussion section.  There are elements of discussion in the results section that are not helpful for the reader.

Specific issues:

1)     Introduction section: line 101:  it is no longer unclear whether the Arg690Cys TMPO variant can cause DCM (refs).  It is clear that it is a functional polymorphism.

2)     Methods section: lines 247-251 seems to be mistaken, please check and clarify.

3)     Results section, line 265: the term “in silico” for mutation classification seems inappropriate as the ACMG criteria include bioinformatic programs but also include other criteria to classify a mutation as pathogenic or likely pathogenic.  Please check if this term was used appropriately in this statement.

4)     Figure 1:  Please add information on the echo and electrocardiograms of mothers who were found to be mutation carriers. 

5)     Figure 1-C: please do not fully fill the square of the sibling with sudden cardiac death, as cardiomyopathy was not confirmed and is misleading.  Fill this square in gray or some other way to clarify this issue.

6)     Were other causes of hypertrophic cardiomyopathy ruled out in patient 1?

7)     Line 365: given current definitions, it is best to use the word “sex” instead of “gender”.

8)     Results section 3.4 (line 399):  was the 59 year-old control used for the dermal fibroblast studies male or female?  Please clarify.

9) According to gnomAD, the frequency of pLOF TMPO variants is not actually high, their frequency is similar to that statistically expected (no constraint). 

10) Section 4.2, line 731:  please use the term sex instead of gender.

11) Please add a short paragraph describing study limitations.

Reviewer 2 Report

This manuscript is a substantial, convincing and important contribution to understanding the molecular mechanisms of nuclear 'laminopathy' diseases. The authors discovered three new rare mutations in TMPO, which encodes multiple protein isoforms of LAP2. Their multi-faceted clinical, cell biological and biochemical results firmly establish the genetic link between TMPO and cardiomyopathy, which was previously uncertain. Two novel mutations specifically affect LAP2-alpha, a protein important for nucleoskeletal support of activities in the nuclear 'interior': missense p.Ala240Thr predicted to affect binding to chromatin protein HMGN5, and frameshift p.Gly395Glufs*, predicted to destabilize the entire protein or delete downstream functional regions. The third novel mutation, pLeu124Phe, affects the highly-conserved LEM-domain, which binds BAF. They also studied a previously-identified mutation, p.Arg690Cys, four residues away from the C-terminus, predicted to affect binding to A-type lamins and/or HMGN5. Their results strongly support the first two predictions, and provide many interesting and unexpected insights including (but not limited to): that LAP2alpha haplo-insufficiency causes clinical disease; that TMPO gene expression is strongly sex-regulated (lower in males, esp. in left ventricle of the heart); and dominance of all three including the p.Arg690Cys-mutated protein, which caused high frequency 'micronucleation' and may provide important new insight into the role of LAP2alpha during exit from mitosis and/or rebuilding nuclear lamina architecture (e.g., see 2004 paper by Dechat, Gajewski, Korbei et al., J. Cell Sci). Their evidence also suggests the reduced level of LAP2alpha protein is being sensed, for positive feedback control of LAP2alpha gene expression. Only minor edits and clarifications, detailed below, are needed.

Clarify abstract lines 60-61, and results lines 669-681: Mentioning known partner Rb will make it easier to follow. E.g., in abstract: "...BAF, and failed to inhibit E2F1 transcription factor activity, consistent with the loss of pRb protein, which is normally stabilized by LAP2alpha." E.g., add to results line 681: "...associate with BAF, since p.Leu124Phe is not predicted to affect binding to pRb."

Figure 1C, Family-III. Comment or explain on line 338 why individual II-3 is shown as homozygously affected (-/-), when the father's genotype is unknown and IL-3's box lacks a question mark, and individual II-3 is both alive and unaffected.

Line 307: Define "NYHA"

Figure 3C, bottom right panel and line 436: The 'delta' symbol is mysterious and undefined in the legend. Change to "truncated?".

Line 548: Cite the relevant reference and state that p.Arg690Cyc is a "previously-identified but molecularly uncharacterized mutation".

Lines 553-554: Separate into two sentences, as follows: "...these micronuclei were positive for lamin A/C and lamin B1 (Figure 6D), [as expected, or not? cite ref?]. Micronuclei are known to indicate mitotic defects [35]."

Lines 574-576: Confusing. Simplify by naming mutants. E.g., "Given their locations outside and upstream of the regions required for LAP2 [alpha?] homodimerization (aa 459-693), binding to pRb (aa 415-615) and binding to lamin A/C (aa 615-693), mutations p.Leu124Phe and p.Ala240Thr were not expected to..".

Line 585-586: Delete "the two missense variants of the C-ter region of LAP2a, the novel"

Line 604: Delete "L124F"-- it is not shown in Figure 7.

Figure 8F, left-hand labels misspelled? "LAP2b lblot" and "HA lblot"

Line 674-675: change to "and the two alpha-specific missense variants p.(Ala240Thr) and p.(Arg690Cys)..."

Line 703, tell the readers exactly which variant is referred to in ref. 21.

Lines 780-781: Double-negative is confusing.

Line 799: Change "The C-ter variants" to "LAP2alpha-specific variants"

Line 805: The HMGN5-binding region is quite large. Has this been mapped more precisely? I.e., is the entire region required, or is a smaller region sufficient to bind HMGN5?
